

# Uncertainties in forecast surface mass balance outweigh uncertainties in basal sliding descriptions for 21st Century mass loss from three major Greenland outlet glaciers.

J. Rachel Carr[1], Emily A. Hill[2], and G. Hilmar Gudmundsson[2]

[1]School of Geography, Politics, and Sociology, Newcastle University, Newcastle-upon-Tyne, NE1 7RU, UK
[2]Department of Geography and Environmental Sciences, Northumbria University, Newcastle-upon-Tyne, NE1 8ST, UK

**Correspondence:** Rachel Carr (rachel.carr@newcastle.ac.uk)

**Abstract.** The Greenland Ice Sheet contributed 10.6 mm to global sea level rise between 1992 and 2018 and is projected to be the largest glacial contributor to sea level rise by 2100. Here, we assess the relative importance of two major sources of uncertainty in 21st century ice loss projections: 1) the choice of sliding law and 2) the surface mass balance (SMB) forecast. Specifically, we used the ice flow model Úa to conduct an ensemble of runs for 48 combinations of sliding law and SMB

forecast for three major Greenland outlet glaciers with differing characteristics (Kangerlussuaq [KG], Humboldt [HU] and Petermann [PG] glaciers) and evaluated how the importance of these uncertainties varied between the study glaciers. Overall, our results show that SMB forecasts were responsible for 4.45 mm of the variability in sea level rise by 2100, compared to 0.33 mm SLE due to sliding law. HU had the largest absolute contribution to sea level rise and the largest range (2.16 to 7.96 mm SLE), followed by PG (0.84 and 5.42 mm SLE), and these glaciers showed similar patterns of ice loss across the SMB

forecasts and sliding laws. KG had the lowest range and absolute values (-0.60 to 3.45 mm SLE) of sea level rise and the magnitude of mass loss by SMB forecast differed markedly from HU and PG. Our results highlight SMB forecasts as a key focus for improving estimates of Greenland's contribution to 21st century sea level rise.

## 1   Introduction

The Greenland Ice Sheet (GrIS) lost approximately 4000 billion tonnes of ice between 1992 and 2018, which equated to a sea

level rise contribution of 10.6 mm (IMBIE, 2020) and it is forecast to be the largest cryospheric contributor to 21st Century sea level rise (Box et al., 2017). Forecasts estimate that the GrIS will contribute $90 \pm 50$ and $32 \pm 17$ mm to sea-level rise for RCP8.5 and RCP2.6, respectively by 2100 (Goelzer et al., 2020) and under certain RCP8.5 scenarios, the contribution may reach 167 mm (Choi et al., 2021). Ice is lost from the GrIS via two main mechanisms, which contribute approximately equally to current losses (Mouginot et al., 2019): changes in surface mass balance ([SMB] i.e. the difference between accumulation

and ablation) and accelerated discharge of ice into the ocean from marine-terminating outlet glaciers (Mouginot et al., 2019; IMBIE, 2020). Both of these mechanisms have contributed to the dramatic increase in ice loss from the GrIS since the 1970s (Mouginot et al., 2019). Forecasts suggest that the dynamic component of ice loss will continue to be significant in the future, accounting for 22-70% of ice loss from Greenland by 2100 (Choi et al., 2021). Ice loss and the sea level rise contribution from





Greenland's marine-terminating outlet glaciers has been highly temporally and spatially variable (e.g. Carr et al., 2017; Howat

and Eddy, 2011; Moon and Joughin, 2008; Murray et al., 2015), with neighbouring glaciers often exhibiting very different

behaviour (Carr et al., 2015; Porter et al., 2014). As such, it is vital to be able to accurately model the dynamic behaviour of

Greenland's outlet glaciers and their potential future contribution to sea level rise.

An important source of uncertainty in projections of future sea level rise

from large ice masses such as GIS is the description of basal motion. This

is usually done in ice-flow models through the use of a basal sliding law,

which relates the stresses acting at the bed to the basal sliding velocity.

However, the mathematical form of this sliding law is subject to consid-

erable uncertainties and each sliding law typically contains some parame-

ters that are poorly constrained. For example, in the often-used Weertman

sliding law, a power-law relationship is assumed between basal drag and

basal velocity, involving two parameters, i.e. the stress exponent ($m$) and

a pre-factor ($C$), that is usually referred to as the basal slipperiness. Both

of these parameters are poorly constrained, and the basal slipperiness ($C$)

likely shows considerable spatial variation. When conducting numerical

modelling studies using Weertman sliding law, we thus need to infer both

the basal slipperiness distribution ($C$) and the rate factor in Glen's flow

law ($A$), with the latter relating the rate of ice deformation to the stress

applied to the ice (Joughin et al., 2019; Nias et al., 2018; Barnes and Gud-

mundsson, 2022; Åkesson et al., 2021).

Basal slipperiness is hard to measure directly and direct observations

are limited, and so it is usually estimated by using an ice sheet model to

invert from measured surface ice velocities. Given the high flow speeds

of Greenland outlet glaciers (Joughin et al., 2010), much of the motion

must occur through basal sliding. This can occur through a variety of

processes, including soft sediment deformation (Blankenship et al., 1987;

Boulton and Jones, 1979); enhanced creep or regelation around obstacles

(Weertman, 1957, 1964); and/or the presence of water at the bed (Iken

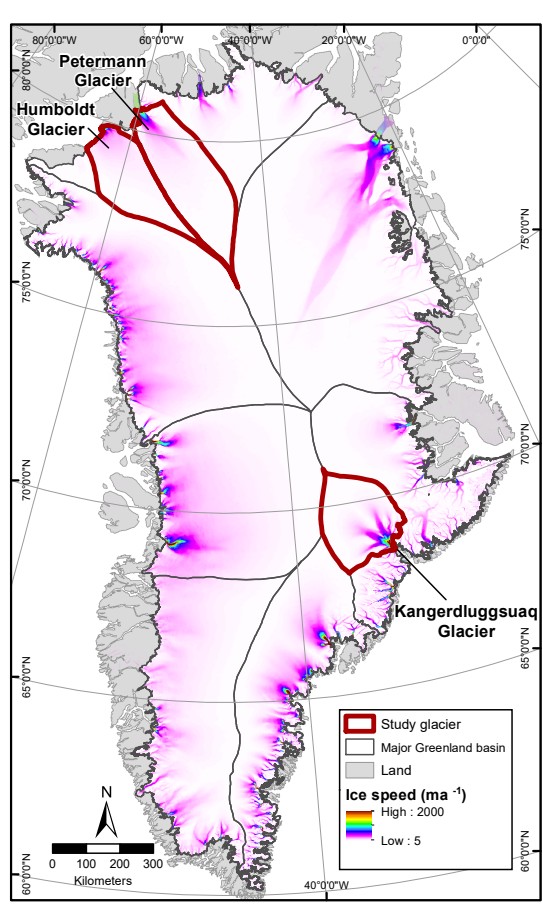

**Figure 1.** Location map, showing the three study glacier catchments, ice speed, and major Greenland outlet glacier sectors

and Bindschadler, 1986; Iken et al., 1983), particularly linked cavities (Lliboutry, 1968). Several different sliding laws have

been defined (e.g. Budd et al., 1979; Tsai et al., 2015; Weertman, 1957) which are described in detail in Section 3.1. However,

most sliding laws are some combination of the Weertman law, i.e. a power law relationship between basal drag ($\tau_b$) and

sliding velocity ($U_b$), so that $u = C\tau^m$ (where $C$ and $m$ are constants) and Coulomb friction, in which basal drag is equal

to effective pressure ($N$), multiplied by a constant ($\mu_k$), so that $\tau = \mu_k N$. Most previous numerical modelling studies have

used the Weertman sliding law, but recent work has shown that the choice of sliding law can significantly impact grounding



line behaviour (Brondex et al., 2017; Gladstone et al., 2017) and estimates of ice loss and sea level rise (Brondex et al., 2019;
Joughin et al., 2019; Åkesson et al., 2021; Lilien et al., 2019), making it crucial to take into account potential differences in transient run outputs resulting from the choice of sliding law.

Work in the Amundsen Sea Embayment assessed the impact of three different sliding laws on ice loss, namely a Weertman, a Schoof and a Budd sliding law (Brondex et al., 2019). Results showed that the Weertman sliding law consistently gave the lowest ice losses, whilst the Budd sliding law always gave the highest (Brondex et al., 2019). Previous work has also
demonstrated that grounding line positions in a synthetic model can vary by 100s of km, dependant on sliding law (Brondex et al., 2017). The impact of sliding law has only been evaluated at one Greenland glacier to date: Petermann Gletsjer (PG), northern Greenland (Åkesson et al., 2021). Here, grounding line retreat differed by 10s of kilometers between sliding laws, whilst the sea level rise contributions varied by 133 % for 2 °C of warming and 282 % for 5 °C (Åkesson et al., 2021). Thus, the choice of sliding law appears to have a major impact on ice dynamics and sea level rise for West Antarctica and PG, but it is
unclear how this varies across the diverse range of outlet glacier characteristics (e.g. ice velocity, terminus type and catchment area) found in Greenland.

A second major source of uncertainty in future projections of sea level rise from ice sheets is the estimates of future SMB under different climate change scenarios (Hofer et al., 2020; Goelzer et al., 2020; Payne et al., 2021; Nowicki et al., 2020). Often SMB is determined using ensemble climate forecasts from global climate models (GCMs), which are down-scaled using
Regional Climate Models (RCMs): these better represent key processes occurring over the GrIS, such as snow proprieties (Hofer et al., 2020). GCM inter-comparison exercises are run to determine the full spread of modelled future climate change, with the latest being the Coupled Model Intercomparison Project Phase 6 (CMIP6; Eyring et al. (2016); Nowicki et al. (2020)). Previous studies have demonstrated that climate model uncertainties contribute markedly to the spread in future sea level rise estimates from the GrIS: they accounted for 36 mm of the spread up to 2100, in the context of a total sea level rise forecast of
between $90 \pm 50$ (RCP8.5) and $32 \pm 17$ mm (RCP2.6; Goelzer et al. (2020)). Consequently, it is vital to assess the impact of different SMB scenarios on forecast sea level rise from Greenland outlet glaciers.

Here we evaluate the impact of four different sliding laws (Section 3.1) and 12 different future SMB scenarios (Section 3.5) on ice loss in sea level equivalent (SLE; mm) from three major GrIS outlet glaciers from 2015 to 2100. We focus on Kangerdluggsuaq Gletsjer (KG), east Greenland, and Sermersuaq (Humboldt) Gletsjer (HU) and PG, northern Greenland
(Fig. 1). These glaciers have differing flow speeds, geometries and terminus types and are located in different regions of the GrIS (Section 2), which enables us to test whether the differences in transient behaviour observed with different sliding laws and SMB scenarios persist between glacier types. They are also major outlets from the GrIS and contribute markedly to its dynamic ice loss, making them important foci for numerical modelling work. We used the same input data for each glacier, from 2014/15, and the same initial model parameters, to ensure comparability. For each glacier, we conducted four inversions,
using the Weertman, Budd, Tsai and Cornford sliding laws, and then ran the model forward in time for 100 years. This was done for each of the 12 SMB scenarios and for a control run, where SMB was fixed at 2014/15 values. Once completed, we assessed differences in sea level rise contribution, according to sliding law, SMB scenario, and study glacier.



## 2 Study glaciers

Our study glaciers were chosen due to their diverse characteristics and their importance for Greenland-wide mass loss: KG
contributed a total of 154.1 Gt to ice loss between 1972 and 2018, making it the third largest contributor, HU was the fourth
largest, contributing 141.0 Gt, and PG was 23rd largest at 56.0 Gt (Mouginot et al., 2019). KG was largely in balance until
2003, after which ice loss accelerated dramatically, so that it accounted for 14 % of the total GrIS discharge anomaly between
2000 and 2012 (Enderlin et al., 2014) and was the largest source of dynamic ice loss in 2018 (Mouginot et al., 2019). HU and
PG have lost mass since 1970s, with the rate of ice loss increasing from the mid-2000s (Mouginot et al., 2019). The study
100 glaciers have different terminus types, calving styles and basal topography. PG has large floating ice tongue, which extends
∼46 km from its grounding line (Nick et al., 2012; Johannessen et al., 2013; Münchow et al., 2016, 2014) and ∼80 % of its
mass loss occurs via basal melting (e.g. Münchow et al., 2014; Rignot et al., 2009; Wilson et al., 2017). PG has shown limited
velocity response to removal of large sections of its ice tongue, either in observations or modelling (Nick et al., 2012; Hill et al.,
2018a, 2021, 2018b; Rückamp et al., 2019). HU's terminus is largely grounded (Carr et al., 2015) and it is the widest glacier in
105 Greenland at ∼90 km (Weidick, 1995). The northern section of HU is underlain by deeper bedrock, which slopes downwards
inland (Carr et al., 2015) and, consequently, it is faster flowing, calves large tabular icebergs and has retreated an order of
magnitude more then at the southern section in recent decades (Carr et al., 2015). Neither KG nor HU have permanent ice
tongues and ice loss is dominated by calving, rather than basal melt. KG is one of the GrIS's fastest flowing glaciers, reaching
almost 10 (km a$^{-1}$) at its terminus, where as maximum speeds at PG and HU are ∼1.3 km a$^{-1}$ (Fig. 1).

## 110 3 Methods

### 3.1 Sliding laws

In ice flow models, basal sliding laws are used to relate basal drag $\boldsymbol{\tau}_b$ to the ice sliding velocity, $\boldsymbol{v}_b$. Ice-flow models typically
use sliding laws where either 1) the functional relationship between basal drag and basal sliding is prescribed, or 2) the basal
drag is prescribed directly, or 3) some combination of 1 and 2. Example of the first type is the Weertman sliding law (Weertman,
115 1957) where

$$\boldsymbol{\tau}_b^W = \beta^2 \boldsymbol{v}_b \tag{1}$$

where the superscript $W$ has been added to indicate that this is the basal drag obtained when using Weertman sliding law, and
where $\beta^2$ is defined as

$$\beta^2 = C^{-1/m} \|\boldsymbol{v}_b\|^{1/m-1}$$



and $C$ is be basal slipperiness, and $m$ a stress exponent. Another example of the first approach is Budd sliding law (Budd et al., 1979), where

$$\boldsymbol{\tau}_b = \beta^2 N \boldsymbol{v}_b \tag{2}$$

where $N$ is the effective basal water pressure. Coulomb sliding law is an example of the second type where the basal drag is defined at each bed location as

$$\boldsymbol{\tau}_b^C = \mu_k N \frac{\boldsymbol{v}_b}{\|\boldsymbol{v}_b\|} \tag{3}$$

where $\mu_k$ is the coefficient of kinetic friction. A commonly used example of the third approach, where Weertman and Coulomb sliding laws are combined using a reciprocal sum as

$$\frac{1}{\|\boldsymbol{t}_b\|^m} = \frac{1}{\|\boldsymbol{t}_b^W\|^m} + \frac{1}{\|\boldsymbol{t}_b^C\|^m}$$

This particular approach of combining Weertman and Coulomb sliding law was first suggested in Asay-Davis et al. (2016), and used in model inter-comparison experiments Cornford et al. (2020), and is termed the Cornford sliding law in this paper. Another suggestion for combining Weertman and Coulomb type sliding laws is using

$$\boldsymbol{\tau}_b = \min(\boldsymbol{\tau_b}^W, \boldsymbol{\tau}_b^C) \tag{4}$$

as proposed by Tsai et al. (2015).

Calculating the effective basal water pressure distribution, $N$, requires a subglacial hydrological model. Here we use a simple model based on the assumption of perfect hydrological connection where

$$N = g\rho(h - h_f)$$

where $h$ is the ice thickness, $h_f$ the thickness at flotation, and $\rho$ the ice density. This assumption of perfect hydrological connection becomes increasingly unrealistic with distance away from the grounding line. This limitation of the hydrological model is unlikely to have significant impact on sliding laws that combine Weertman and Coulomb type behaviour, as the effective pressure is only of relevance in regions where the Coulomb drag is less than the Weertman basal drag. However, for Budd sliding law, the effective pressure is used within the whole domain and the limitations of the hydrological model can be expected to impact modelled ice flow, as discussed in Section 4.3.

### 3.2 Model set-up

We used the numerical model Úa (Gudmundsson et al., 2012), which is a vertically integrated ice flow model that uses the shallow ice stream / shelf approximation to solve the ice dynamic equations (MacAyeal, 1989). Úa uses a finite element approach and has an adaptive mesh, allowing key areas to be resolved in greater detail, as required. Úa has been previously





used to determine the dynamic response of Greenland outlet glaciers to ice tongue loss (Hill et al., 2021, 2018b) and to investigate the impact of ice shelf loss on outlet glacier behaviour in Antarctica (De Rydt et al., 2015; Reese et al., 2018; Gudmundsson et al., 2023).

For each of our study glaciers we generated a triangular finite element mesh (Figures S1-S3). For each glacier, the inland boundaries were determined from ice surface velocity and elevation data and the boundary at the termini was identified using Landsat 8 imagery from 2014 (Fig. 1). Meshes were generated using the Mesh2d unstructured mesh generator (Engwirda, 2014). The total number of nodes and elements for each mesh were as follows: KG (49,073 elements; 24,841 nodes), HU (89,205 elements; 177,545 nodes), PG (55,634 elements; 28,293 nodes). At KG and HG, the mesh was refined based on ice

velocity and effective strain rate, whilst PG's mesh was refined based on distance to the grounding line, ice velocity and surface elevation. At KG and HG, the maximum element size was 4 km (ice velocity <50 m a$^{-1}$) and a minimum of 0.08 km at KG (ice velocities >4000 m a$^{-1}$) and 0.07 km at HG (ice velocities >400 m a$^{-1}$). At PG, element size was a maximum of 15 km (surface elevation >1,200 m.a.s.l. and ice velocities <10 m a$^{-1}$) and a minimum of 0.3 km over the ice tongue (surface elevation <750 m a.s.l. and ice velocities >250 m a$^{-1}$). For all three glaciers, the inland catchment boundaries were set to zero and fixed

i.e. had no inward, outward or parallel flow and the termini had natural boundary conditions. PG has a floating ice tongue and nunataks, so required additional boundary conditions: velocities were fixed at zero along the lateral margins of PGs tongue and nunataks had velocities fixed to zero in the normal and tangential directions (Hill et al., 2021, 2018b).

We used a number of remotely sensed datasets to initialise the ice flow model and the same datasets were used for all three modelled domains, to ensure consistency. We chose the nominal date of winter 2014/15, as this is the earliest date at

which the MEaSUREs Greenland Ice Sheet Velocity Map from InSAR Data, Version 2 data were available at 200 m resolution (Joughin et al., 2020). This enabled us to use the highest resolution velocity data available, whilst enabling us to compare our modelled grounding line fluxes to measured (Table S2). Furthermore, 2015 marked the start of the SMB projections. The glacier geometries were determined from BedMachine v3 (Morlighem et al., 2017), which has a spatial resolution of 150 m and is derived from a combination of observations and mass conservation. Specifically, we utilised the ice surface, ice thickness,

glacier bed geometry and the offshore bathymetry. Annual surface mass balance (SMB) was used to initialise the transient runs and was sourced from RACMO v2.3 (Noël et al., 2016). We used the average SMB for the years 2013-2017, to ensure that the SMB was representative and not overly biased by an individual year. For KG and HU, no basal melting was applied, as neither glacier has a substantial floating section. At PG, we used our model-optimised velocities to calculate sub-shelf melt rates from ice flux divergence and these melt rates were then fixed during all forward-in-time experiments.

**3.3     Model inversion**

Using inverse methodology, as described in (Hill et al., 2021, 2018b; Gudmundsson et al., 2019) we simultaneously inverted for the rate factor in Glen's Flow Law ($A$) and basal slipperiness ($C$). Following convention, $n$ in Glen's Flow Law and the parameter $m$ were set to 3. The inversion minimises the cost function $J$ between the observed ($u_{\mathrm{obs}}$) and modelled ($u_{\mathrm{mod}}$) velocities, where the cost function is the sum of a misfit ($I$) and regularisation ($R$) term. Where the misfit term is $I = I_u + I_{\dot{h}}$



and are defined as

$$I_u = \frac{1}{2\mathcal{A}} \int (u_{\mathrm{mod}} - u_{\mathrm{obs}})^2 / \epsilon_{\mathrm{obs}}^2 \, d\mathcal{A} \tag{5}$$

and

$$I_{\dot{h}} = \frac{1}{2\mathcal{A}} \int (\dot{h}_{\mathrm{mod}} - \dot{h}_{\mathrm{obs}})^2 / \epsilon_{\mathrm{obs}}^2 \, d\mathcal{A} \tag{6}$$

where $\dot{h}$ is the rate of thickness change, $\epsilon_{\mathrm{obs}}$ are the measurement errors, and the total area of the computational domain is $\mathcal{A} = \int d\mathcal{A}$. The regularisation term is defined as

$$R = \frac{1}{2\mathcal{A}} \int \left( \gamma_s^2 (\nabla \log_{10}(p/\hat{p}))^2 + \gamma_a^2 (\nabla \log_{10}(p/\hat{p}))^2 \right) d\mathcal{A} \tag{7}$$

where $\hat{p}$ are the a prior values for model parameters ($\hat{A}$ and $\hat{C}$). It can be shown that this regularisation is equivalent of assuming the prior distributions of $A$ and $C$ to be described by the Matérn family of covariance functions (Whittle, 1954; Lindgren et al., 2011). The gradients of $J$ with respect to $A$ and $C$ were determined using the adjoint method and Tikonov regularisation was applied to the $A$ and $C$ fields. Tikhonov regularisation parameters $\gamma_s$ and $\gamma_a$ control the slope and amplitude of the gradients in $A$ and $C$. Optimum values were determined using $L$-curve analysis and are equal to $\gamma_s = 10000$ and $\gamma_a = 1$ for all results presented. An example $L-$curve is shown in Figure S4.

We penalised the rate of thickness change using observations of surface elevation change $\dot{h}_{\mathrm{obs}}$ taken from the 'Greenland SEC grid from Cryosat-2 dataset' (see Simonsen and Sørensen (2017) for further details) . The data were provided as two-year means and we used values for 2014-15, to correspond with our surface velocities. The dataset was provided with error estimates, but these were very small and might unduly contain the fit to ice velocities. Thus, we tested a range of error values $\epsilon_{\mathrm{obs}}^2$ for the rate of surface elevation change, specifically: error multiplied by 1 (mean error = 0.047 m a$^{-1}$), 2 (mean error = 0.094 m a$^{-1}$), 5 (mean error = 0.235 m a$^{-1}$) and 10 (mean error = 0.47 m a$^{-1}$), and no dhdt used in the inversion. We then examined the velocity misfit for each inverse run and chose dhdt errors x10, as this minimised the velocity misfit but also enabled us to include errors in dhdt in the inversions (Table S1).

A separate inversion was run for each sliding law, i.e. Weertman, Budd, Tsai and Cornford (Table 1 and Fig. S5) and for each glacier, so that 12 $A$ and $C$ fields were produced in total. Inversions were run until they had fully converged (Fig. S6): the model had fully converged after 16,000 iterations at HU, after 30,000 iterations at KG and after 10,000 iterations at PG (Fig. S6). For all three glaciers, the mean difference between measured and modelled velocities was comparable between sliding laws (Table 1 and Fig. S5). For KG, HU and PG, the $C$ fields are broadly similar for the Weertman, Tsai and Cornford sliding laws and are markedly different for Budd, reflecting the fact that $C$ in the Budd law has different dimensions to $C$ in the other sliding laws (Fig. S7). Similarly, the spatial patterns and absolute values for $A$ were comparable between the Weertman, Tsai and Cornford sliding laws for all three glaciers (Fig. S8). For the Budd sliding law, values of $A$ are somewhat higher at KG and lower at PG, compared to the other sliding laws (Fig. S8).





**Table 1.** Mean difference between modelled and measured ice velocities at the end of the inversion (m a⁻¹), for Kangerdluggsuaq Gletsjer (KG), Humboldt Gletsjer (HU) and Petermann Gletsjer (PG). Mean differences are given for each sliding law

|  | Weertman (W) | Budd (W-N0) | Tsai (minCW-N0) | Cornford (rCW-N0) |
|---|---|---|---|---|
| HU | 15.40 | 16.92 | 15.41 | 15.47 |
| KG | 3.87 | 5.68 | 3.58 | 3.61 |
| PG | 10.75 | 12.04 | 10.84 | 10.76 |

### 3.4  Relaxation and initial state

Our model, similar to others, experiences a period of model drift at the beginning of a forward-in-time simulation, characterised by a thickening and slowdown of the fast-flowing glaciers. This is due to inconsistencies in the input datasets and uncertainties in model parameters. To relax our model, and bring the mass loss trend into line with present-day observations, we iteratively applied a correction to the mass balance term $a_s$. We subtracted the misfit between modelled and observed rates of thickness change ($\dot{h}$), determined from Cryosat-2 two-year mean data for 2014-2015 (Simonsen and Sørensen, 2017), from the RACMO surface mass balance field ($a_s^0$) as follows

$$a_s^N = a_s^{N-1} + |\Delta a_N| \tag{8}$$

where

$$|\Delta a_N| = \left.\frac{dh}{dt}\right|_{\text{obs}} - \left.\frac{dh}{dt}\right|_{N-1} \tag{9}$$

We initialised our model at the year 2010 and, at each run step, $N$ subtracted the previous modelled thickness change $\left.\frac{dh}{dt}\right|_{N-1}$ from observations $\left.\frac{dh}{dt}\right|_{\text{obs}}$. After five years, we found that the difference between observed and modelled $\frac{dh}{dt}$ had converged to within the error of the observations (Fig. S9). Specifically, modelled $\frac{dh}{dt}$ was within the measured $\frac{dh}{dt}$ error at KG and within two times the $\frac{dh}{dt}$ measurement error at HU and PG (Fig. S9). For all three glaciers, modelled $\frac{dh}{dt}$ was well within ten times the $\frac{dh}{dt}$ estimated measurement error, which was used in the inversion (Fig. S9). For each glacier, we repeated this correction for all four sliding laws (Fig. S9). These final corrected mass balance fields are then kept fixed for all forward simulations and perturbations are applied as anomalies on-top of this (see Section 3.5).

### 3.5  Forward Experiments

To estimate the impact the impacts of sliding law and SMB scenario on projected sea level rise up to 2100, we ran forward simulations for four sliding laws (Weertman, Budd, Tsai and Cornford) and 12 SMB scenarios, plus a control run for each sliding law with no SMB forcing. For each sliding law, an inversion was conducted to estimate the $A$ and $C$ for the respective sliding laws. For each glacier, we used the same input geometry, finite element mesh and boundary conditions, to ensure consistency between experiments. To generate control runs, we conducted a series of forward transient runs with the SMB forcing kept constant using the 2014/15 SMB for each sliding law. Next, to evaluate the impacts of different SMB scenarios



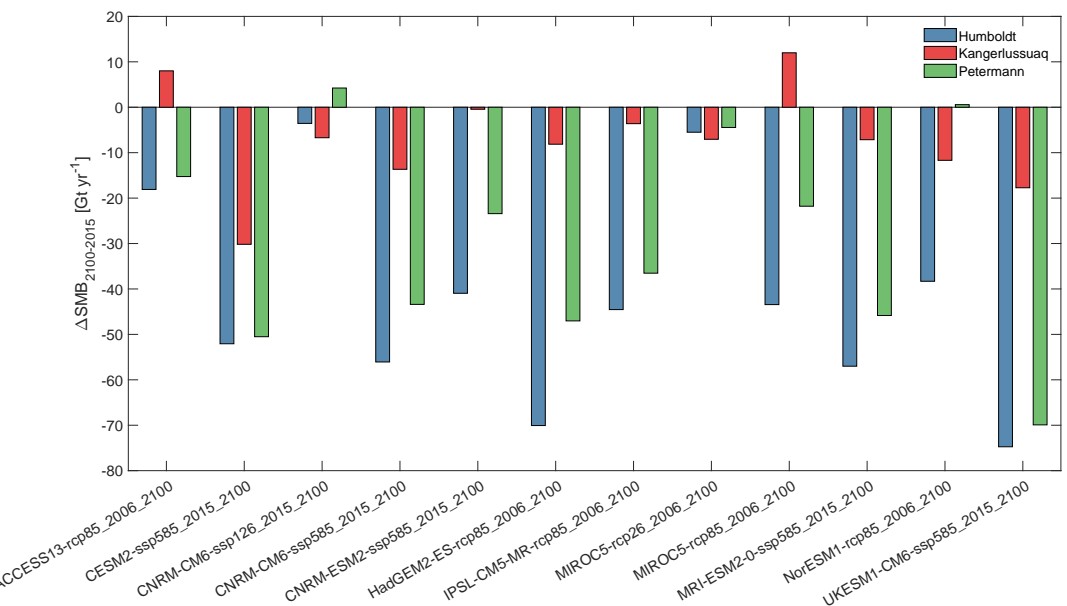

**Figure 2.** Bar graph of the projected surface mass balance (SMB) change between 2015 and 2100, integrated across the catchment of each study glacier. Projections are given for each of the downscaled SMB forecasts and change in integrated surface mass balance is given for Humboldt Gletsjer (blue), Kangerlussuaq Gletsjer (red) and Petermann Gletsjer (green).

on ice losses, we applied a series of SMB scenarios, which projected SMB from 2015 to 2100 (Fig. 2). The 12 scenarios

were downscaled from CMIP5 and CMIP6 GCM forecasts, using the regional climate model MAR (Hofer et al., 2020). The magnitude of projected SMB change integrated for the entire catchment and its variability between the study glaciers is shown in Figure 2. The spatial distribution of each SMB forecast for each glacier is shown in Figures S10, S11, S12 and is presented in Section 4.2.1. In total, this gave us a matrix of 12 SMB scenarios and one control run, for four different sliding laws and for three different glaciers, totalling 156 experiments. For each experiment, we calculated sea level rise equivalent. Each transient

run was run for 85 years, from 2015 to 2100, and had an initial timestep of 0.01 years. Adaptive time stepping was used, whereby the time step is reduced if the number of non-linear iterations in the previous five timesteps exceeds the target of 4 non-linear iterations. We did not use adaptive mesh refinement during the forward runs. The calving front was kept fixed and ice thickness was reset to 1 m if it reduced below 1 m.

## 4   Results

**4.1   Ice loss projections**

Our results show that the combined sea level rise contribution of our three study glaciers by 2100 ranges between 2.40 mm and 16.83 mm, dependent on SMB scenario and sliding law (Figs. 3 and 4). HU has the run with the highest forecast contribution



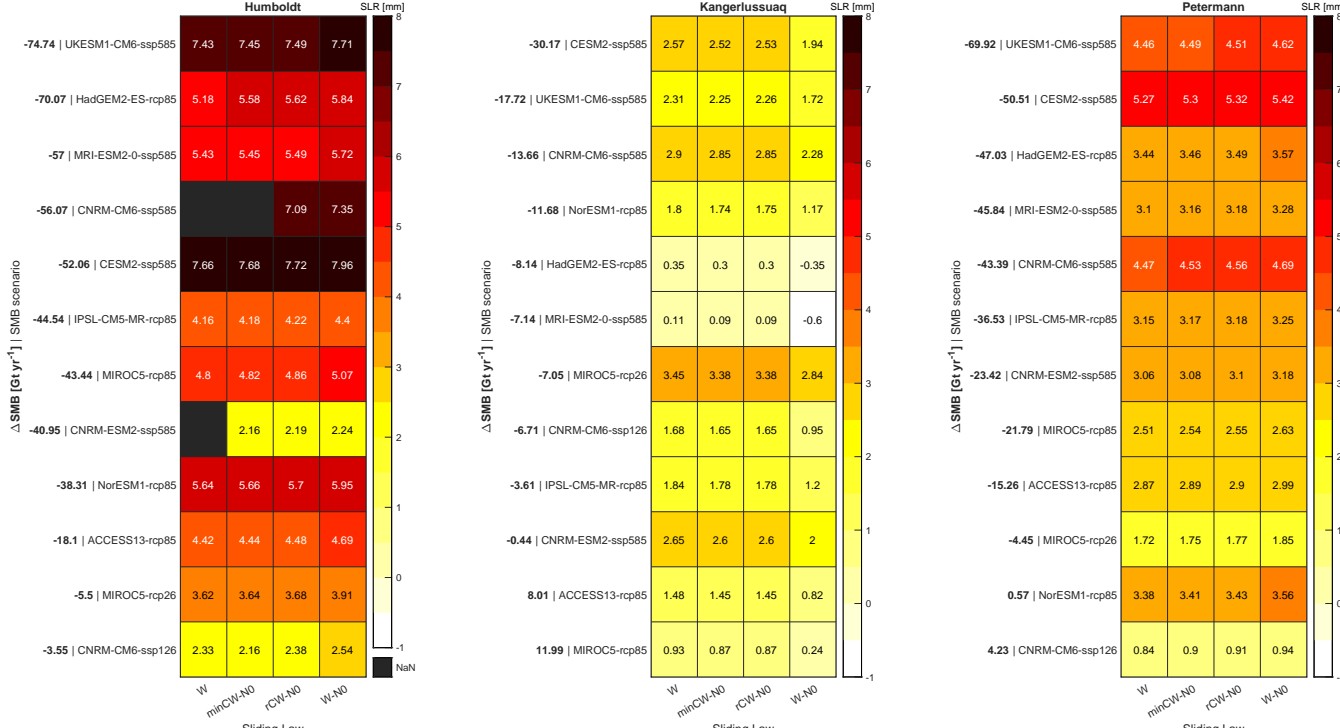

**Figure 3.** Heat map showing the contribution to sea level for each combination of sliding law and surface mass balance (SMB) forecast between 2015 and 2100, for Humboldt Gletsjer (left panel), Kangerlussuaq Gletsjer (middle panel) and Petermann Gletsjer (right panel). Squares are coloured coded according to forecast sea level rise contribution, with dark red colours and positive values indicating the greatest net contribution to sea level rise and white colours and negative values indicating a net reduction in the contribution to sea level rise. The SMB forcing scenarios are given on each $y$-axis and are ordered based on the smallest to largest change in the integrated SMB across each catchment for each scenario (the value is given in bold).

to sea level rise at 7.96 mm by 2100, which occurred with the SMB scenario CESM2-ssp585 and the Budd sliding law (Figs. 3a and 4a). HU's minimum sea level rise contribution by 2100 was 2.16 mm, which occurred for CNRM-ESM2-ssp585

(Weertman and Tsai sliding laws) and for CNRM-CM6-ssp126 (Tsai) (Fig. 4a). PG had the second highest contribution to sea level rise, at 5.42 mm, which was for the run with CESM-ssp585 and the Budd sliding law (Figs. 3b and 4b). PG's lowest sea level rise contribution (0.84 mm) came from CNRM-CM6-ssp126 with the Weertman sliding law (Figs. 3b and 4b). KG had the lowest contribution to sea level rise, reaching a maximum of 3.45 mm, from scenario MIROC5-rcp26 and the Weertman sliding law (Figs. 3c and 4c). Two scenarios at KG gave a net reduction in sea level rise contribution, reaching up to 0.60

mm for HadGEM2-ES-rcp85, for the Budd sliding law (Figs. 3c and 4c). Average grounding line retreat across all runs was smallest at KG (∼5 km), followed by PG (∼10 km) and HU (∼20 km).



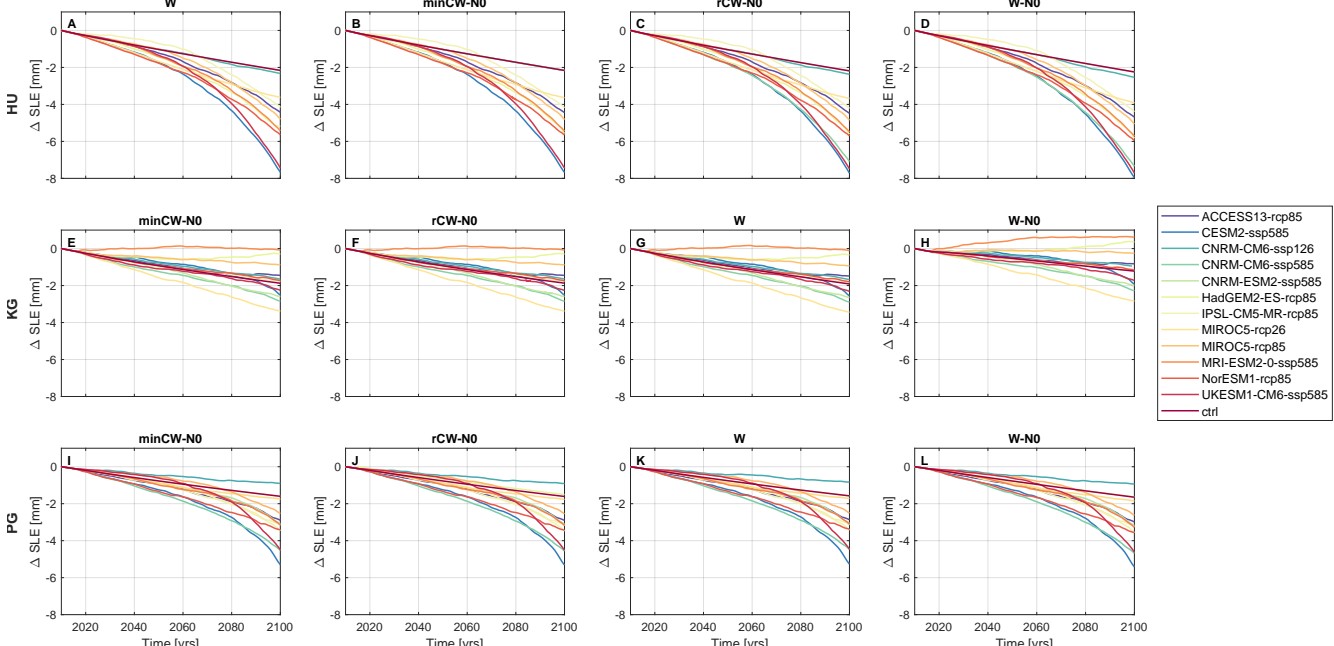

**Figure 4.** Change in sea level rise equivalent between 2015 and 2100, by sliding law and surface mass balance forecast (SMB), for Humboldt Gletsjer (HU; top panel); Kangerlussuaq Gletsjer (KG; middle panel) and Petermann Gletsjer (PG; bottom panel). Plots are divided by sliding law: minCW-N0 = Tsai; rCW-N0 = Cornford; W = Weertman; and W-N0 = Budd. The results are colour-coded by SMB forecast and are in alphabetical order.

## 4.2 Surface mass balance scenarios

### 4.2.1 Variability in surface mass balance scenarios

The magnitude and spatial distribution of forecast changes in integrated SMB between 2015 and 2100 varied markedly between
the three study glaciers and the different scenarios. At HU and PG, the change in integrated SMB was comparable between SMB scenarios (i.e. the scenario that produced the greatest SMB change at HU also resulted in the greatest change at PG) but the magnitude of that change was greater at HU (Fig. 2). This pattern did not persist at KG and the magnitude of SMB change between 2015 and 2100 was much lower in all scenarios than at HU and PG, with the glacier undergoing a net gain in mass in two scenarios (ACCESS13-rcp85 and MIROC5-rcp85; Fig. 2). HU's SMB became more negative by 2100 over almost the
entire catchment for all RCP8.5 scenarios, and changes were particularly marked at lower elevations (Fig. S10). For the two RCP2.6 scenarios, SMB became slightly more negative over the lower third of the catchment and showed limited change inland (Fig. S10). For the majority of the RCP8.5 forecasts at KG, SMB became more negative between 2015 and 2100 in the area ~100 km inland of the terminus, particularly on the eastern side of the catchment, and SMB showed limited change further inland (Fig. S11). The change in SMB was most negative for the HADGEM2-ES-rcp85 scenario and reached furthest inland





for CESM2-SSP585 (Fig. S11). Compared to the other RCP8.5 scenarios, the change in SMB between 2015 and 2100 was less negative and less extensive for the scenarios ACCESS13-rcp85 and MIROC5-rcp85, resulting in limited net change in SMB (Fig. S11 and Fig. 2). RCP2.6 scenarios at KG showed a similar spatial pattern to HU, whereby SMB became slightly more negative between 2015 and 2100 near the terminus and change was limited inland (Fig. S11). At PG, SMB generally became more negative within ∼240 km of the terminus for the RCP8.5 scenarios, with the remainder of the catchment showing little

change (Fig. S12). Compared to the other RCP8.5 scenarios at PG, the trend to a more negative SMB extended furthest inland for CESM2-SSP585 and UKESM1-CM6-ssp585 and was smaller in magnitude for ACCESS13-rcp85, CNRM-ESM2-ssp85 and MIROC5-rcp85, (Fig. S12). As at HU and KG, RCP2.6 scenarios showed more limited change than for RCP8.5 and areas of more negative mass balance were only evident within ∼125 km of the terminus (Fig. S12).

### 4.2.2    Variability in sea level rise contribution by surface mass balance scenario

Our results demonstrate that the choice of SMB scenario was responsible for the majority of variability in forecast sea level rise for all three study glaciers (Fig. 3 and 4): averaged across the three glaciers, the range in sea level rise contribution across the SMB forecasts was 4.45 mm, compared to 0.33 mm across the sliding laws. This difference persists for both low (RCP2.6) and high (RCP8.5) emissions scenarios: RCP 2.6 and 8.5 scenarios are available for the MIROC5 and CNRM-CM6 scenarios and in each case, the variability between sliding laws for a given SMB scenario is less than that between SMB forecasts with

the same RCP emissions scenarios (e.g. MIROC5 2.6 versus CNRM-CM6 2.6; Fig. 3 and 5). For all three study glaciers, the variability in grounding line retreat between SMB scenarios was limited (Figs. S13, S14 and S15).

        HU and PG have a similar sea level rise contribution depending on the emission scenario, although the absolute values for HU are larger: HU has both the run with the largest contribution to sea level rise (7.96 mm; CESM2-ssp585) and the greatest range in sea level rise contribution (5.80 mm; Figs. 3a, 4a and 5a). For comparison, the range in sea level rise values

at PG was 4.58 mm, with a maximum of 5.42 mm (CESM2-ssp585; Figs. 3b, 4b and 5b). KG had the lowest contribution to sea level rise, reaching a maximum of 3.45 mm (MIROC5-rcp26) and the smallest range in values (4.05 mm). Averaged across all sliding laws, the highest sea level rise contribution at HU was given by the forecast CESM2-ssp585, followed by UKESM1-CM6-ssp585and CNRM-CM6-ssp585 (Fig. 3a). At PG, CESM2-ssp585 gave the highest average values, followed by CNRM-CM6-ssp585 and UKESM1-CM6-ssp585(Fig. 3b). Thus, we observed commonalities in the SMB scenarios and

sliding law combinations that produced the greatest contribution to sea level rise at HU and PG (Fig. 3a and b and 5a and b). In contrast, at KG, MIROC5-rcp26 gave the highest sea level rise contribution across the sliding laws (Figs. 3c, 4c and 5c), but was a low-range scenario at HU and PG (Figs. 3, 4 and 5). SMB scenario CNRM-CM6-ssp126 gave the lowest sea level rise contributions at both HU and PG (Fig. 3). However, the SMB forecasts giving the lowest contribution varied between glacier, specifically: at HU (CNRM-ESM2-ssp585 and CNRM-CM6-ssp126), at KG (MRI-ESM2-0-ssp585 and HadGEM2-ES-rcp85)

and at PG (CNRM-CM6-ssp126 and MIROC5-rcp26). Overall, there was some commonality in which SMB forecasts resulted in the highest sea level rise contribution at HU and PG, but KG differed substantially from the other two glaciers, and for all three glaciers there was a large range in potential sea level rise contribution between the full range of SMB scenarios (Figs. 3 and 4).



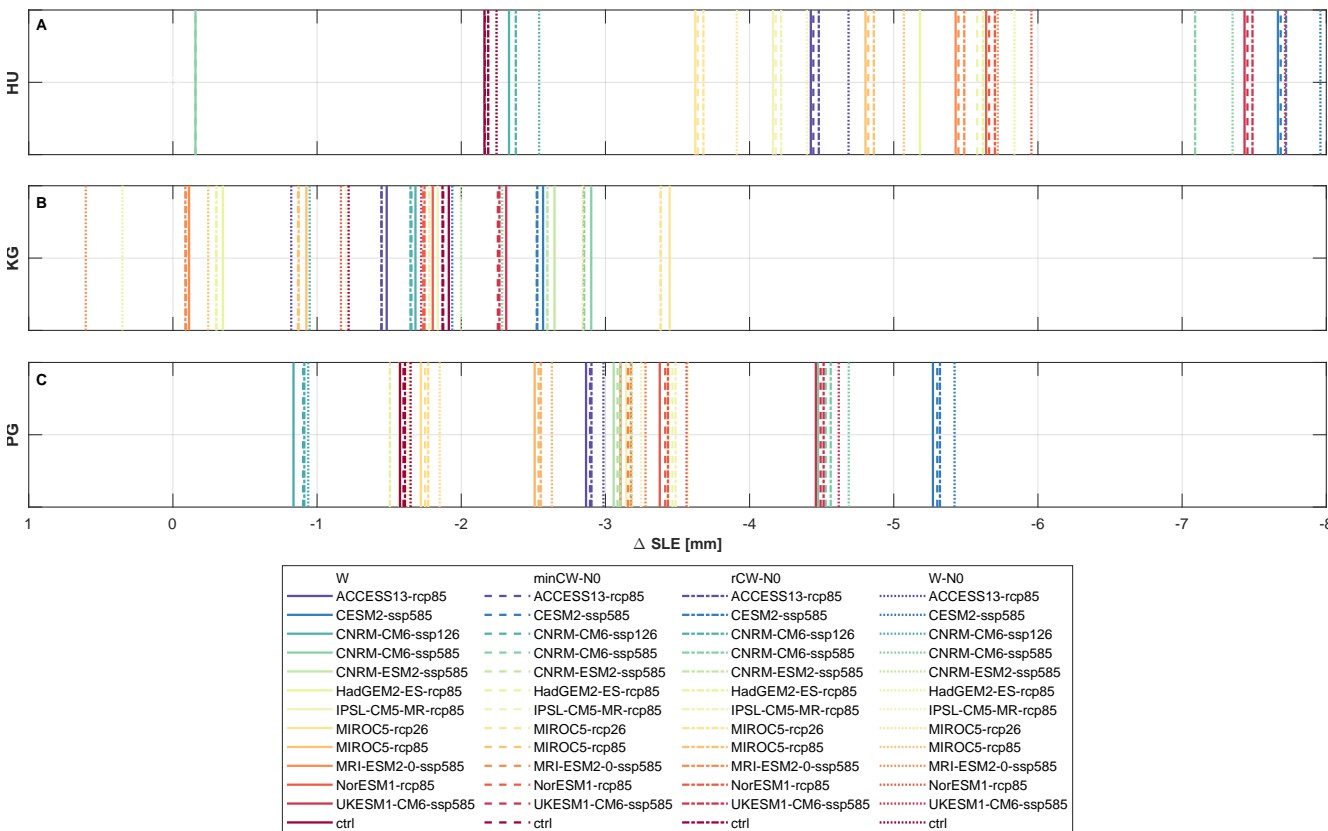

**Figure 5.** Bar graph of sea level rise contribution for Humboldt Gletsjer (HU; top panel); Kangerlussuaq Gletsjer (KG; middle panel) and Petermann Gletsjer (PG; bottom panel). Sliding laws are symbolised according to line style (W = –; minCW-N0 = – –; rCW-N0 = -.; W-N0 = ..) and SMB forecasts by colour. The sliding laws are: W = Weertman; minCW-N0 = Tsai; rCW-N0 = Cornford; and W-N0 = Budd.

## 4.3 Sliding laws

Overall, our results show that the choice of sliding law had limited impact on the sea level rise contribution from our three study glaciers and that the majority of the variability was due to the choice of SMB forecast (Figs. 3, 6, 7 and 8). The impact of sliding law was also not consistent between glaciers: at HU and PG, Budd consistently gave the highest contribution to sea level rise and Weertman was generally lowest. Conversely, at KG, Weertman was consistently the highest and Budd was consistently lowest, with Budd resulting in mass gain for certain SMB scenarios (HadGEM-es-rcp85 and MRI-ESM2-0-ssp585; Figs. 3,

6, 7 and 8). Tsai and Cornford generally gave intermediate values at all three glaciers. The differences in 2100 grounding line position between each sliding law were small for all study glaciers (Figs. S13, S14 and S15).

At HU, Weertman, Tsai and Cornford produced similar results, with Budd giving slightly greater mass loss (Fig. 6). However, the differences were small, regardless of RCP scenario: below we use MIROC5 to exemplify this, as it has both an RCP2.6 and RCP 8.5 scenario. For MIROC-rcp85, the difference between the lowest sea level rise contribution (Weertman: delta SLE =



4.80 mm) and the highest (Budd: delta SLE = 5.07 mm) was 0.27 mm at 100 years (Fig. 6). This difference between Budd and Weertman persisted for low emissions scenarios, with MIROC-rcp26 differing by 0.29 mm between Budd and Weertman (Fig. 6). At PG, the differences between sliding laws was even smaller than at HU, e.g. for MIROC-rcp85, the difference in sea level rise contribution between Budd and Weertman was 0.12 mm and for MIROC-rcp2.6 it was 0.13 mm (Figs. 3b and 8). At KG, the differences between the sliding laws was greater than at HU and PG (Figs. 3c and 7). For example, for MIROC5-rcp26, the

difference between Budd and Weertman was 0.61 mm and for MIROC5-rcp85 it was 0.69 mm (Figs. 3c and 7). Differences between the other sliding laws at KG were minimal (Figs. 3c and 7). Thus, with the exception of the Budd sliding law at KG, the choice of sliding law resulted in limited differences in the forecast sea level rise by 2100 for our study glaciers.

## 5   Discussion

### 5.1   Ice loss projections

Forecasts for our three study glaciers give a total sea level rise contribution by 2100 that is comparable to that from the entire GrIS between 1992 and 2018: estimates suggested the GrIS added $10.8 \pm 0.9$ mm to global sea level between 1992 and 2018 (IMBIE, 2020), whilst our mean contribution by 2100 was 9.53 mm and a maximum of 16.83 mm (Fig. 3). Focusing on future projections, previous work has applied the same SMB scenarios used here to the entire GrIS, as part of the Ice Sheet Model Intercomparison Project for CMIP6 (ISMIP6; Goelzer et al. (2020)). This study forecast sea level rise contributions of $90 \pm 50$

and $32 \pm 17$ mm to sea-level rise for RCP8.5 and RCP2.6, respectively (Goelzer et al., 2020). For comparison, our projections estimated mean sea level contributions of 10.52 mm (RCP8.5) and 6.74 mm (RCP2.6) from the three study glaciers (Fig. 3), highlighting their importance for future losses at the GrIS scale. Subsequent numerical modelling work, using CMIP5 RCP8.5 and CMIP6 SSP585 scenarios, forecast a higher Greenland-wide contribution to sea level rise, at 79 to 167 mm by 2100 (Choi et al., 2021) and suggested that ice dynamics would account for between 22 and 70 % of ice loss (Choi et al., 2021). Even

under these higher-end estimates, our forecasts for HU, KG and PG equate to 5-10 % of Greenland's sea level rise contribution for our mean values (9.53 mm) and 10-20 % for our most extreme scenarios (16.83 mm; Fig. 3). It should be noted that calving and SMB elevation feedbacks are not included in our model runs, so these estimates should be treated as low-end.

   HU had the highest contribution to sea level rise by 2100 (maximum: 7.96 mm; mean: 4.75 mm), PG was second highest (maximum: 5.42 mm; mean: 3.12 mm) and KG was much lower (maximum: 3.45; mean: 1.66 mm) (Figs. 3 and 5). Thus,

the relative contribution of each study glacier to Greenland's 21st Century sea level rise differs from contemporary patterns: between 1972 and 2018 KG was the third highest contributor to cumulative ice loss (154.1 Gt), Humboldt was the fourth (141.0 Gt) and Petermann was 23rd (56.0 Gt) (Mouginot et al., 2019). This increase in HU's relative importance highlights it as a key site for future research, particularly as it has received much less scientific attention than KG (e.g. Luckman et al., 2006; Howat et al., 2008, 2007; Joughin et al., 2008). We attribute HU's high ice loss to the strongly negative integrated SMB

change observed for most of the SMB scenarios between 2015 and 2100 (Fig. 2). Our calculated range of sea level rise values are slightly lower than those in a recent study (5.2-8.7 mm; Hillebrand et al. (2022)), most likely because our work does not



**Figure 6.** Change in sea level rise equivalent between 2015 and 2100, by sliding law and surface mass balance forecast (SMB), for Humboldt Gletsjer (HU). Plots are divided by SMB forecast, given in the subplot title. The results are coloured-coded by sliding law: minCW-N0 = Tsai; rCW-N0 = Cornford; W = Weertman; and W-N0 = Budd.



**Figure 7.** Change in sea level rise equivalent between 2015 and 2100, by sliding law and surface mass balance forecast (SMB), for Kangerlus-suaq Gletsjer (KG). Plots are divided by SMB forecast, given in the subplot title. The results are coloured-coded by sliding law: minCW-N0 = Tsai; rCW-N0 = Cornford; W = Weertman; and W-N0 = Budd.



**Figure 8.** Change in sea level rise equivalent between 2015 and 2100, by sliding law and surface mass balance forecast (SMB), for Petermann Gletsjer (PG). Plots are divided by SMB forecast, given in the subplot title. The results are coloured-coded by sliding law: minCW-N0 = Tsai; rCW-N0 = Cornford; W = Weertman; and W-N0 = Budd.





include calving and instead focuses on SMB. As such, future work should repeat our SMB scenarios runs, but with calving added, to determine the full range of likely sea level rise contribution from HU by 2100.

PG's contribution to sea level rise between 2015 and 2100 averaged 3.12 mm and ranged between 0.84 and 5.42 mm (Figs.
3b and 5b). These values are markedly higher than the forecast sea level rise contribution from a range of scenarios of ice tongue calving and enhanced basal melt: even the most extreme scenario, i.e. total loss of PGs tongue and enhanced basal melt, generated only 0.92 mm of sea level rise by 2100 (Hill et al., 2021). This suggests that SMB is a major contributor to both absolute ice loss at PG and the uncertainty in that ice loss. This is supported by previous flow-line modelling, which suggested that SMB was responsible for almost all ice loss from PG up to 2100, with dynamic losses only becoming significant
in the late 22nd Century (Nick et al., 2013). Furthermore, both previous numerical modelling (Hill et al., 2018b; Nick et al., 2010, 2012; Rückamp et al., 2019) and remote sensing studies (Johannessen et al., 2013; Hill et al., 2018a) demonstrated PG's limited response to large calving events. This limited response has been attributed to PG's comparatively weak tongue (Hill et al., 2018b; Nick et al., 2012) and the presence of a topographic high ~12 km inland of its 2020 position, which, along with a narrowing fjord, is likely to suppress future grounding line retreat (Hill et al., 2021). PG has high basal melt
rates beneath its ice tongue (35 ma$^{-1}$; Rignot and Steffen (2008)), which account for a large portion of its contemporary mass loss (Rignot et al., 2009; Münchow et al., 2014; Wilson et al., 2017). However, applying melt rates at the top end of recent observations (50 m$^{-1}$; Wilson et al. (2017)) resulted in considerably less sea level rise contribution (0.65 mm with no change in ice tongue extent; Hill et al. (2021)) than all of our SMB experiments (Figs. 3b and 5b). It should be noted, however, that our basal melt rates were fixed at these upper-end contemporary estimates, due to uncertainties in potential future changes, and
so basal melt rates, and their respective contribution to ice loss, at PG may well increase in the future. Finally, recent work has investigated the conditions required to re-grow PG's ice tongue, once lost (Åkesson et al., 2022), and even major changes to SMB were insufficient to re-grow the tongue: re-growth was only possible with large reductions in iceberg calving and ocean cooling (Åkesson et al., 2022). As such, the relative importance of SMB versus calving in driving changes in sea level rise contributions from PG may differ, depending on whether the scenario involves warming and ice loss, versus cooling and ice
growth (Åkesson et al., 2022).

KG had the lowest contribution to sea level rise of our study glaciers, which we attribute to SMB forecasts being less negative, or even positive, over its catchment, compared to HU and PG (Fig. 2 and Fig. S2). Previous flow-line modelling suggested that KG would undergo a net gain in SMB during the 21st and 22nd Centuries and attributed all ice loss to changes in dynamics, rather than SMB (Nick et al., 2013). Similarly, flow-line modelling up to 2065 suggested that RCP8.5 forcing had
limited impact on KGs ice front position (Barnett et al., 2022). Our results, generated from a 2-HD model, suggest that SMB does influence KG's sea level rise contribution and forecasts net ice loss overall for the vast majority of runs (Fig. 3c), but this is offset by KG's comparatively positive SMB during the 21st Century (Fig. 2 and S2). This is in agreement with results from Greenland-wide forecasts of ice loss under the MIROC5 RCP8.5 scenario, which suggest that KG's high rates of dynamic losses during the 21st Century are offset by a generally positive SMB in south-east Greenland (Choi et al., 2021). Observational
data has demonstrated that KG's dynamics, and hence its contribution to sea level rise, have varied dramatically over monthly to decadal time scales (e.g. Luckman et al., 2006; Howat et al., 2008, 2007; Joughin et al., 2008; Bevan et al., 2012; Khan et al.,





2014)). KG's terminus is now located at its most retreated position on record, which has brought it to the edge of a substantial basal overdeepening (Bevan et al., 2019; Brough et al., 2019). However, there is uncertainty over whether KG will undergo rapid retreat and ice loss in the near future, induced by positive feedbacks associated with traversing an overdeepening (e.g.

Meier and Post, 1987; Schoof, 2007), or whether the positive SMB observed in our data (Fig. 2 and 3c) and in previous work (Choi et al., 2021) is sufficient to offset dynamical effects during the 21st Century. Results from Greenland-wide simulations suggest that positive SMB will enable KG to maintain its current frontal position (Choi et al., 2021) and previous work has shown that retreat into an overdeepening does not necessarily equate to rapid dynamic loss (Gudmundsson et al., 2012). However, our data show a substantial range in KG's ice loss (Fig. 5c and 3c), due to differences in forecast SMB (Fig. 2), and

this variability makes it unclear whether SMB will indeed compensate for dynamic losses during the 21st Century. Given KG's capacity to rapidly generate major discharge anomalies (Enderlin et al., 2014) and uncertainties over its future behaviour, we highlight it as a priority for future numerical modeling work.

## 5.2  Surface mass balance scenarios

The choice of SMB scenario resulted in much larger variability in 21st Century sea level rise contribution (4.45 mm) than the

choice of sliding law (0.33 mm) for our study glaciers (Figs. 3 and 5). This pattern persisted between RCP8.5 and RCP2.6 scenarios (Figs. 3 and 5) and we also attribute the differing sea level rise forecasts from our study glaciers to differences in patterns of forecast SMB (Fig. 2). Specifically, HU consistently lost the most ice (Fig. 3a) and had the most negative forecast SMB across the 12 scenarios (Fig. 2), with negative SMB extending across most of its catchment in all RCP8.5 scenarios (Fig. S10). Our sea level rise contribution from PG was slightly lower than at HU (Fig. 3b) and so was its integrated forecast

SMB (Fig. 2), whilst negative SMB was confined to a smaller proportion of PG's catchment (Figs. S10, S12). Overall, HU and PG showed a similar pattern in integrated SMB across the SMB scenarios, e.g. scenarios that produced more negative (positive) SMB at HU also produced more negative (positive) SMB at PG (Fig. 2), most likely due to their spatial proximity (Fig. 1). In contrast, KG underwent the least ice loss and its integrated forecast SMB was far less negative than at HU and PG for all scenarios, with some scenarios predicting positive SMB (Fig. 2). This is consistent with forecasts of largely positive

SMB during the 21st Century at KG (Choi et al., 2021). Furthermore, the pattern of integrated SMB values between the different scenarios at KG showed little resemblance to that at HU and PG (Fig. 2). Thus, SMB forecasts may be comparatively consistent within regions of the GrIS, but may vary substantially between them (Fig. 2 and Nowicki et al. (2020)), leading to notable differences in forecast ice loss from major outlet glaciers.

Previous work evaluated the performance of CMIP5 GCMS for both Greenland and Antarctica, relative to contemporary

reanalysis and gridded observational data and 21st Century forecasts of climate and oceanic change (Barthel et al., 2020). Results suggested that HadGEM2-ES, MIROC5, and NorESM1-M were most appropriate for Greenland, for the purposes of ISMIP6 (Barthel et al., 2020). However, even focusing on these forecasts, the mean range in sea level rise for our study glaciers is 1.37 mm (Fig. 3 and 5), highlighting the need for further work on identifying the most appropriate SMB forecasts to use at both the ice sheet scale and for specific outlet glaciers and regions. This needs also needs to be assessed with CMIP6 forecasts,

as Barthel et al. (2020) focused just on CMIP5 and studies suggest that mass loss from SMB varies substantially between



CMIP5 and CMIP6 models (Choi et al., 2021; Payne et al., 2021). Furthermore, previous work has highlighted differences in spatial patterns of SMB between HadGEM2-ES, MIROC5, and NorESM1-M (Nowicki et al., 2020). Based on currently available data and SMB forecasts, we therefore cannot confidently rule out any of our ensemble predictions of sea level rise.

### 5.3 Sliding laws

Our results show that the choice of sliding law had limited effect on the sea level rise contribution from our study glaciers (Figs. 6, 7 and 8). This broadly agrees with results from synthetic experiments focused on Weertman, Budd and Schoof sliding laws (Brondex et al., 2017), which showed that the difference in volume above flotation (VAF) was small and comparable to our results, for Weertman (1-3%) and Schoof (1-5%), whilst the Budd sliding law accounted for the majority of the variability (15-29%; Brondex et al. (2017)).

Significant variability in VAF and grounding line retreat has been observed in some Antarctic simulations, which have primarily focused on the ASE (Joughin et al., 2019; Nias et al., 2018; Brondex et al., 2019; Lilien et al., 2019; Barnes and Gudmundsson, 2022). For example, using a Schoof law resulted in 24 mm of additional sea level rise from the ASE, compared to a Weertman law, although the impact of the sliding law was sensitive to the viscosity during the inversion (Brondex et al., 2019). We suggest that this difference in sensitivity to sliding laws between our study glaciers and those in the ASE reflects

their diffing characteristics. First, our study glacier catchments are an order of magnitude smaller than the ASE and have much narrower termini that are constrained by fjord walls (Fig. 1): e.g. KG's terminus is approximately 5 km wide and PG's is 20 km, compared to 40 km at Pine Island. Lateral resistive stress scales inversely with glacier width (Raymond, 1996), meaning that lateral stresses are likely to be more important for Greenland outlet glaciers, thus reducing their sensitivity to changes in basal slipperiness associated with different sliding laws. Furthermore, the Budd sliding law assumes a perfect hydrological

connection with the ocean, which would have a greater impact on glaciers with larger catchments. The ASE is backed by a major overdeepening (e.g. Morlighem et al., 2020; Fretwell et al., 2013; Rignot et al., 2014), which has the potential to rapidly generate feedbacks between grounding-line retreat and ice loss, following an initial retreat. In turn, this could lead to greater variability between sliding laws, if certain laws move the grounding lines past key pinning points. Whilst our study glaciers do have overdeepenings inland (e.g. Carr et al., 2015; Bevan et al., 2019; Morlighem et al., 2017; Brough et al., 2019), they

are much smaller-scale than those in the ASE and our modelled grounding line retreats (Figs. S13, S14 and S15) were limited compared to those reported for the ASE (Joughin et al., 2019; Brondex et al., 2019; Nias et al., 2018; Lilien et al., 2019). Thus, smaller overdeepenings and more limited grounding line retreat may reduce the possibility of unstable behaviour and hence reduce the variability in sea level rise contribution between the sliding laws.

For HU and PG, our results show that the Budd sliding law consistently produced the highest sea level rise contribution

and Weertman was the lowest (Fig. 3a and b). This agrees with previous work at PG (Åkesson et al., 2021) and from the ASE (Brondex et al., 2019; Barnes and Gudmundsson, 2022). The variation in sea level rise contribution results from the differing physics of each sliding law and, in particular, the role of effective pressure. As ice thins (e.g. due to dynamic changes or negative SMB), effective pressure drops, and, in the Budd sliding law, this results in a linear decrease in basal shear stress (see Equation 2). Effective pressure is not included in the Weertman law (see Equation 1), meaning there is no link between effective



pressure and basal shear stress, whilst the Tsai and Cornford only include effective pressure when certain conditions are met, usually close to the grounding line, and a Weertman approach is applied elsewhere (Equations 3 and 4). As noted above, the Budd law (Equation 2) assumes a perfect hydrological connection, meaning that effective pressure can be comparatively low far inland, which enables dynamic changes to propagate further inland than for other laws (Brondex et al., 2017, 2019; Åkesson et al., 2021; Barnes and Gudmundsson, 2022). The assumption of perfect connectivity is likely to be reasonable close to the grounding line, as per the Tsai and Cornford laws, but unlikely to hold far inland, as in the Budd law (Barnes and Gudmundsson, 2022). As such, results to date (Figs. 3 and 5; Åkesson et al. (2021); Barnes and Gudmundsson (2022)) suggest that Weertman laws usually provide a lower bound for sea level rise estimates and Budd an upper bound. However, at KG, our results produce slightly lower sea level rise estimates for Budd compared to the other sliding laws, which we attribute to the greater sensitivity of sliding velocities to changes in ice thickness when using Budd sliding law. Thus, we suggest that the Budd sliding law may not be physically realistic and sea level rise forecasts using this law are likely to be outliers (Barnes and Gudmundsson, 2022).

Our results show far more limited variation between sliding laws (Fig. 3b and 5b) than recent work at PG (Åkesson et al., 2021): our range in sea level rise contribution for PG across the four sliding laws averaged 0.13 mm by 2100, compared to 2.15 mm for 5 °C warming and 0.99 mm for 2 °C warming (Åkesson et al., 2021). We suggest that this is at least partly due to the differing choice of sliding laws: Åkesson et al. (2021) use a Weertman law, then a Budd law, a Schoof law, and three types of Coulomb-type till-friction law (Åkesson et al., 2021). In contrast, we use a Weertman law, a Budd law and then two laws (Tsai and Conford) that switch between a Weertman-type and Coulomb-type friction law, depending on effective pressure. It may be that these 'mixed' friction laws are more realistic, but data are not available to confirm if this is the case, due to the inaccessibility of glacier beds beneath fast-flowing outlet glaciers (Barnes and Gudmundsson, 2022). Another potential reason for the observed difference is that our study does not include calving and Åkesson et al. (2021) note that the variability in contribution to sea level rise is much lower in ocean-only warming scenarios. Calving is a major source of uncertainty in modelling Greenland outlet glaciers (Goelzer et al., 2020), and so an important next step is to assess how different calving implementations effect variability in sea level rise, in combination with different sliding laws and SMB forecasts. Previous numerical modelling work suggested that PG's contribution to 21st Century sea level rise may be limited by the presence of a bedrock ridge ∼12 km inland of its current terminus (Hill et al., 2021). However, for certain sliding laws (Budd and till-assimilation-N-Budd), previous studies have been able to retreat PG's grounding line past this bedrock ridge (Åkesson et al., 2021). As such, initially small differences in grounding line retreat, resulting from the choice of sliding law, could result in major variations in sea level rise contribution, dependant on the detailed bed topography. Similarly, results from the ASE demonstrated that accurate representation of the bedrock had at least as much impact of sea level rise forecasts as the choice of sliding (Nias et al., 2018). Thus, having accurate and detailed data on bed topography, particularly near the grounding line, is vital for forecasting ice loss from Greenland outlet glaciers.

Overall, our results suggest that the impact of sliding laws on sea level rise contribution and grounding line retreat is limited for three major Greenland outlet glaciers (Figs. 3, S13, S14, and S15). This agrees with some results from the ASE (Barnes and Gudmundsson, 2022), but contrasts with recent findings at PG (Åkesson et al., 2021). Thus, we recommend assessing





the impact of sliding laws on sea level rise estimates at a greater number of Greenland outlet glaciers to confirm or refute
our findings. Furthermore, the impact of the sliding laws varies with the forcing applied, including SMB scenarios (Fig.
3), and ocean melt and calving (Åkesson et al., 2021; Barnes and Gudmundsson, 2022). Thus, future work should take an
ensemble approach, to identify the largest source(s) of uncertainty, but this is challenging, given that many key factors are
hard to parameterise (e.g. calving and basal melt rates) and the parameter space would be very large. At the ice sheet scale,
certain SMB forecasts may provide a better fit to observations (Barthel et al., 2020) and certain spatial patterns may persist
across multiple scenarios (Nowicki et al., 2020), but we cannot rule out any of our sea level rise forecasts, based on available
data. Thus, further work is need to determine the most appropriate SMB forecast(s) for a range of different Greenland outlet
glaciers, located in different regions of the ice sheet, given the large regional variability in SMB forecasts (Nowicki et al. (2020)
Figs. S10, S11 and S12). Furthermore, our study does not include feedbacks between SMB and elevation, which should be
incorporated into future work. Ideally results should be evaluated against multi-decadal observations of glacier dynamics, but
there are few glaciers with such records (Åkesson et al., 2021). Future work needs to be conducted at the scale of individual
glaciers to regions, as assumptions and parameterisations at the ice sheet scale are unlikely to be appropriate for individual
outlets (Åkesson et al., 2021; Hillebrand et al., 2022). This work is vital for accurately forecasting Greenland's 21st Century
contribution to sea level rise and for reducing uncertainties associated with outlet glacier behaviour.

## 6   Conclusions

Overall, our results showed that SMB forecasts accounted for the majority of the variability in 21st Century sea level rise from
three major Greenland outlet glaciers, namely Humboldt Gletscher (HU), Kangerlussuaq Gletscher (KG) and and Petermann
Glestcher (PG). Sliding laws had a much smaller impact on sea level rise forecasts, accounting for an average of 0.33 mm in
variability between runs, compared to 4.45 mm between SMB scenarios. Forecast 21st Century sea level contributions from
HU, KG and PG are important at the ice sheet scale and are comparable to the observed total ice loss from the GrIS between
1992 and 2018. HU contributed most to sea level rise, which we attribute to the extensive negative SMB forecast across its
catchment during the 21st Century. PG had the second highest ice losses and variability in SMB scenarios followed a similar
pattern to HU, due to their spatial proximity. Our results suggest that SMB may have a greater influence on 21st Century ice
loss at PG than contemporary basal melt rates and iceberg calving. KG had the lowest sea level rise contribution of our study
glaciers, as its SMB forecasts were much less negative than at HU and PG. Given the range in our predicted ice loss at KG, it
is unclear whether this more positive SMB will offset loses due to ice dynamics and/or enable KG to maintain its grounding
line position during the 21st Century. Integrated SMB values between scenarios differed markedly between KG and HU and
PG, suggesting it may not be appropriate to use a single scenario for glaciers located in different regions of the GrIS. The
choice of sliding law had much less impact on sea level rise contributions than in the Amundsen Sea Embayment, which we
attribute to the differing catchment geometry and subglacial topography of Greenland outlet glaciers. Our results suggest that
the Weertman sliding law usually provides the lower bound for sea level rise forecasts and Budd provides the upper bound, but
the latter may not be physically realistic, due to its dependence on effective pressure across the entire catchment. Future work

should extend our ensemble approach, to include different calving parameterisations and basal melting forecasts. Overall, our findings highlight SMB forecasts as a key source of uncertainty in the response of Greenland's outlet glaciers to climate change

and their potential 21st Century contribution to sea level rise.

*Code and data availability.*

The model runs were conducted using Úa, which is publicly available via GitHub: https://github.com/GHilmarG/UaSource. We used version 2022 for our experiments.

Input datasets used in our experiments were:

– MEaSUREs Greenland Ice Sheet Velocity Map from InSAR Data, Version 2, 200 m resoltuion (Joughin et al., 2020)

– BedMachine v3 (Morlighem et al., 2017)

– RACMO v2.3 (Noël et al., 2016)

– Surface mass balance scenarios from CMIP5 and CMIP6 GCM forecasts, downscaled using the regional climate model MAR (Hofer et al., 2020).

– Surface elevation data from the 'Greenland SEC grid from Cryosat-2 dataset' (Simonsen and Sørensen, 2017).

*Author contributions.*

JRC and EH ran the model simulations. JRC lead the writing, with and EH and GHG contributing substantially. All three authors were involved in devising the experiments and editing the manuscript.

*Competing interests.*

The authors declare that there are no competing interests.

*Acknowledgements.* We acknowledge Newcastle University's Rocket HPC cluster, which was provided free of charge to conduct the model runs.



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
