# Peer review of "Sensitivity to forecast surface mass balance outweighs sensitivity to basal sliding descriptions for 21st Century mass loss from three major Greenland outlet glaciers."

_EGUsphere, 2023_

## Referee Comment (RC2)

Review of *'Uncertainties in forecast surface mass balance outweigh uncertainties in basal sliding descriptions for 21st Century mass loss from three major Greenland outlet glaciers.'*
**By Carr et al., 2023**

**Summary**

This modeling study investigates the relative impact of a combination of four sliding laws and 12 future surface mass balance scenarios on mass loss from three major Greenland outlet glaciers through 2100. The four sliding laws are selected to include a variety of common modeling applications — these include sliding laws that prescribe the relationship between basal drag and sliding (Weertman and Budd), one where basal drag is directly prescribed (Tsai, in the manuscript), and a combination of the two (Cornford). The three glacier experiments are initialized with the same input data and model parameters. The authors find that the choice of SMB scenario imparts far greater variability on projected mass loss than choice of sliding law for all three study glaciers. By contrast, the SMB scenario used had minimal impact on the magnitude of grounding line retreat, which occurred for all three glaciers in the study. The authors note similar behavior from the two northern outlets, Petermann and Humboldt, with greatest mass losses projected using the Budd sliding law. Interestingly, Kangerdlugssuaq (SE Greenland) exhibited, on average, the smallest net mass losses, which were greatest when implementing the Weertman sliding law and an RCP 2.6 SMB scenario. In addition to suggesting a relatively minimized impact of sliding law choice compared to future SMB scenario, study results also emphasize the likely future importance of Humboldt for Greenland wide mass loss, as this catchment showed the highest SLR contribution of nearly 8mm, without considering additional mass losses from elevation feedbacks or dynamic thinning and terminus change.

This manuscript is clearly written and presents important glaciological work, as the impact of sliding choice is quite understudied for Greenland outlet glaciers. The results will add value to the community and add insight to the relative importance of remaining unknowns to future mass loss scenarios. The methods are overall sound and well justified, and clearly described in the main text and supplement. With minor revisions, this manuscript will be suitable for publication in TC. Some figures may be condensed or combined if manuscript length is of concern. I have included my main comments and requests for clarification first, followed by a few minor comments and edits.

**Comments and requests for clarification**

I understand the motivation for holding terminus fronts static and neglecting elevation/dynamic thinning feedbacks for these experiments, given the focus on SMB+sliding law and high computational load. However, I wonder how these exclusions impact the interpretation of grounding line retreat, given that HU and KG, assumed to terminate at the grounding line (no or negligible floating extensions), both show retreated grounding lines while holding initial 2015 fronts stationary. Wouldn't this create an artificial floating ice tongue by the end of the simulations?

In the discussion, reduced SLR projections stemming from the Budd sliding law at KG s attributed to enhanced sensitivity to changing ice thickness. This is suggested to indicate that the use of the Budd sliding law results in unrealistic SLR projections. Given the sensitivity to effective pressures and related basal shear stress in implementing Budd sliding law, does it not also seem reasonable that for KG, where some SMB scenarios resulting in net thickening (SMB gains), this would then manifest in reduced mass losses in Budd sliding law relative to the other sliding laws?

I may have missed this reference in the text, but what is the cause of NaN values for several Weertman SMB runs for Humboldt Glacier (see Figure 3)?

**Line 170**
*"Annual surface mass balance (SMB) was used to initialise the transient runs and was sourced from RACMO v2.3 (Noël et al., 2016). We used the average SMB for the years 2013-2017…"*

Can you please explain the use of RACMO for initialization, but use of MAR for downscaling CMIP5 and CMIP6 for the SMB scenarios?

**Line 214**
*"We subtracted the misfit between modelled and observed rates of thickness change ($\dot{h}$), determined from Cryosat-2 two-year mean data for 2014-2015 (Simonsen and Sørensen, 2017)…"*

Does this indicate that the misfit was computed only at a single time step (here the 2 year mean thickness change during 2014-2015)? Or that this 2-year period serves as a reference elevation for deriving thickness changes? If the latter, can you clarify the period over which dh/dt relative to the reference period were computed?  2010-2015?

**Line 252 through 255**
Figure references indicate 3b showing Pg results, and Figure 3c showing results from KG. I believe these need to be swapped.

**On BedMachine v3**
Why do the authors elect to use BedMachine version 3 over a newer version? Is the topography similar at the three study catchments for version 3 and the newer version of BedMachine?

**Line 296**
Please double-check to ensure figure references and subplot references correspond to the correct glacier.

**Suggestions for Figures 6—8**
Consider using a combination of solid, dotted, and dashed lines for figures that are stacked in a way that all curves appear in the figure. As shown, only W-N0 and rCW-N0 are readable in many subplots, and it is unclear which of these scenarios are overlapping the other two.

**Minor Comments**

Consider adding labels to subplots in Figure 3 if they are referenced by alphabetic notation in the text ("Figure 3c" for example).

Table 1 – Consider adding in paratheses what percentage of mean basin velocity is represented by the misfit value.

Figure 4 – consider an alternate color ramp or alternating line styles and thicknesses. The yellow and light green colors are challenging to see and discern from one another in the figures.

---

## Author Comment (AC1)

**Reviewer 2**

This modeling study investigates the relative impact of a combination of four sliding laws and 12 future surface mass balance scenarios on mass loss from three major Greenland outlet glaciers through 2100. The four sliding laws are selected to include a variety of common modeling applications — these include sliding laws that prescribe the relationship between basal drag and sliding (Weertman and Budd), one where basal drag is directly prescribed (Tsai, in the manuscript), and a combination of the two (Cornford). The three glacier experiments are initialized with the same input data and model parameters. The authors find that the choice of SMB scenario imparts far greater variability on projected mass loss than choice of sliding law for all three study glaciers. By contrast, the SMB scenario used had minimal impact on the magnitude of grounding line retreat, which occurred for all three glaciers in the study. The authors note similar behavior from the two northern outlets, Petermann and Humboldt, with greatest mass losses projected using the Budd sliding law. Interestingly, Kangerdlugssuaq (SE Greenland) exhibited, on average, the smallest net mass losses, which were greatest when implementing the Weertman sliding law and an RCP 2.6 SMB scenario. In addition to suggesting a relatively minimized impact of sliding law choice compared to future SMB scenario, study results also emphasize the likely future importance of Humboldt for Greenland wide mass loss, as this catchment showed the highest SLR contribution of nearly 8mm, without considering additional mass losses from elevation feedbacks or dynamic thinning and terminus change.

This manuscript is clearly written and presents important glaciological work, as the impact of sliding choice is quite understudied for Greenland outlet glaciers. The results will add value to the community and add insight to the relative importance of remaining unknowns to future mass loss scenarios. The methods are overall sound and well justified, and clearly described in the main text and supplement. With minor revisions, this manuscript will be suitable for publication in TC. Some figures may be condensed or combined if manuscript length is of concern. I have included my main comments and requests for clarification first, followed by a few minor comments and edits.

**Comments and requests for clarification**

I understand the motivation for holding terminus fronts static and neglecting elevation/dynamic thinning feedbacks for these experiments, given the focus on SMB+sliding law and high computational load. However, I wonder how these exclusions impact the interpretation of grounding line retreat, given that HU and KG, assumed to terminate at the grounding line (no or negligible floating extensions), both show retreated grounding lines while holding initial 2015 fronts stationary. Wouldn't this create an artificial floating ice tongue by the end of the simulations? In the discussion, reduced SLR projections stemming from the Budd sliding law at KG s attributed to enhanced sensitivity to changing ice thickness. This is suggested to indicate that the use of the Budd sliding law results in unrealistic SLR projections. Given the sensitivity to effective pressures and related basal shear stress in implementing Budd sliding law, does it not also seem reasonable that for KG, where some SMB scenarios resulting in net thickening (SMB gains), this would then manifest in reduced mass losses in Budd sliding law relative to the other sliding laws? In response to the question about the calving fronts, we prescribe a natural boundary condition and do not include calving: as the reviewer notes this is to separate out the impacts of SMB and sliding law and for computational efficiency. As noted by the reviewer, this does create a small floating section, but this section was always small and comparable to the width of HU / KG. In future work, we plan to quantify the relative impact of calving, implemented via a range of different laws. As noted by the reviewer, the Budd sliding law results in reduced mass losses compared to other sliding laws at KG. We note that the application of Budd sliding law, as used here and in several similar studies in that past, is somewhat questionable as it depends on the effective pressure changes throughout the whole computational domain, and not just in the vicinity of the grounding line. Despite our own reservations about this sliding law, we have nevertheless included it as it is a widely known sliding law and has been used in numerous studies in the past.

I may have missed this reference in the text, but what is the cause of NaN values for several Weertman SMB runs for Humboldt Glacier (see Figure 3)? There are 3 runs with NaN, which we realised just prior to submissions we needed to re-run. We will include these values in the updated version.

Line 170 "Annual surface mass balance (SMB) was used to initialise the transient runs and was sourced from RACMO v2.3 (Noël et al., 2016). We used the average SMB for the years 2013-2017…" Can you please explain the use of RACMO for initialization, but use of MAR for downscaling CMIP5 and CMIP6 for the SMB scenarios? We used RACMO for the initialisation as we believed it was marginally better at representing contemporary Greenland surface mass balance and initially, we were only going to test the impact of sliding laws, not SMB forecasts, so there would have been no inconsistency. The downscaling of the CMIP5 and CMIP6 scenarios with MAR was done as part of another study (Hofer et al., 2020), and, importantly, was done for 13 different scenarios, which enabled us to test the impact of that range of SMB scenarios. Through the inversion and initialisation process, we ensured that our initial model state closely matched observed ice velocities and surface elevation change, so that we were confident our initial model set up matched well with observations at the start of the forward runs (i.e. 2015). Thus, any difference in initial model state due to the choice of SMB product is unlikely to have impacted our results and would be very much smaller than the differences we observed between the various SMB forecasts.

Line 214 "We subtracted the misfit between modelled and observed rates of thickness change ( ̇h), determined from Cryosat-2 two-year mean data for 2014-2015 (Simonsen and Sørensen, 2017)…" Does this indicate that the misfit was computed only at a single time step (here the 2 year mean thickness change during 2014-2015)? Or that this 2-year period serves as a reference elevation for deriving thickness changes? If the latter, can you clarify the period over which dh/dt relative to the reference period were computed? 2010-2015? Yes, this is correct that we use a single timestamp of thickness change: the 2-year mean thickness change during 2014-2015 serves as a reference. However, the misfit between modelled and these observed rates of thickness change is calculated iteratively at every run step for a period of 5 years, nominally between 2010 and 2015. We will update the text to make this clearer to the reader.

Line 252 through 255 Figure references indicate 3b showing Pg results, and Figure 3c showing results from KG. I believe these need to be swapped. Apologies, we will update this.

On BedMachine v3 Why do the authors elect to use BedMachine version 3 over a newer version? Is the topography similar at the three study catchments for version 3 and the newer version of BedMachine? We selected BedMachine v3 because BedMachine 4 was not available at the time when we conducted the model runs. The topography for the study glaciers is similar between the versions and it would take months to re-run the experiments for BedMachine4, so this is unfortunately beyond the scope of this paper. In future work, it would be very interesting to assess the impact of different bed products and/or different versions of the bed. However, we are confident that the swap to BedMachine v4 alone is unlikely to result in major differences in our results.

Line 296 Please double-check to ensure figure references and subplot references correspond to the correct glacier. We will double check this in the revised manuscript.

Suggestions for Figures 6—8 Consider using a combination of solid, dotted, and dashed lines for figures that are stacked in a way that all curves appear in the figure. As shown, only W-N0 and rCW-N0 are readable in many subplots, and it is unclear which of these scenarios are overlapping the other two. We will change the line format for these figures, but the key message from these figures is actually that most of the lines do overlap, i.e. that sliding law makes little difference to the sea level rise contribution.

Minor Comments

Consider adding labels to subplots in Figure 3 if they are referenced by alphabetic notation in the text ("Figure 3c" for example). We will add these as requested.

Table 1 – Consider adding in paratheses what percentage of mean basin velocity is represented by the misfit value. We will add these as requested.

Figure 4 – consider an alternate color ramp or alternating line styles and thicknesses. The yellow and light green colors are challenging to see and discern from one another in the figures. We will review the line styles and colours in this figure.

---

## Author Comment (AC2)

**Reviewer 1 (Steph Cornford)**

Carr et al describe numerical simulations of three major Greenland glaciers from the present day to 2100. For each glacier an ensemble of simulations is constructed by varying (a) the climate (via the surface mass balance) according to published future projections, and (b) the physics approximation of sliding at the ice bed. All simulations begin with ice thickness and velocity in line with present day observations. They conclude that their variation in sliding physics has a far lower impact on future sea level rise than their variation in SMB. I would argue that this is model sensitivity rather than uncertainty, without observational calibration of the evolution, but the same language of uncertainty crops up elsewhere. We are happy to replace uncertainty with sensitivity if required.

The experiments make sense as modelling studies, and the conclusions are justified (although differing from one paper, which is discussed) but an important sliding law has been omitted. This is the regularized Coulomb law (e.g Joughin 2019), which is related to the Schoof and Tsai rules that are discussed, but produces Coulomb-like sliding over a wider region of a typical glacier. It might make little difference, but should be considered. It is particularly important because it agrees most with time-dependent satellite observations in some regions (which is also worth noting in the introduction). We thank the reviewer for highlighting this law, which was very recently implemented in Ùa and was therefore not used in the original study. It would take a considerable amount of time to re-run all of our experiments with this additional law, so we feel that its inclusion is beyond the scope of the current paper. As the reviewer notes, as it may well make little difference: one of the key messages of our paper is that sliding law choice generally makes very little difference to sea level rise and we already include the Tsai law, so we do not expect significantly different results with the Joughin law. However, we will add this sliding law to the introduction, so that the reader is aware of it and note it as an avenue for future work to explore.

(Hilmar: This remark by the reviewer is very interesting and this has led us to implement in Ua the sliding law suggested by Joughin. While we have not done new runs for Greenland, we did inversions for WAIS where we have a large library of existing inversion products obtained with a number of different sliding laws. We found, as noted by Joughin, that this sliding law does not automatically result in a reduction of the basal drag to zero as flotation is approached from upstream direction, even in areas of high sliding velocities. For this reason, it appears that Joughin introduced some further modifications to the sliding law parameter to taper the drag down to zero in those areas. While we are definitely interested in exploring the implications of this sliding law of Joughin further, we feel that some additional criterion is required for how the sliding law parameters are modified as the grounding line is approached. Without a clear description of how this is to be done generally, the sliding law of Joughin is arguably not fully defined. We note that in the 'Cornford' sliding law (see comment below) this reduction to zero drag in the vicinity of the grounding line automatically achieved")

I understood the paper in general although there are a small number of typos/grammar errors.

**Specific comments**

Fig 3 is a good figure, but the labels are small. We will increase the size of the labels.

Figures 5,6,7,8 repeat the same data shown in Fig 4. It is not obvious to me that they serve a purpose. At the same time, other useful figures, e.g plots of the model initial state vs observations are not included. We will review these figures and remove them and/or move to the supplementary info.

Please do not name a 'Cornford' sliding law (line 90, 130 and elsewhere). It appears in earlier work (which is cited in both the Cornford and Asay-Davies papers mentioned), and Cornford certainly makes no claim to it . It will also be useful to say something about the physical meaning of each law when they are introduced. We will add a couple of sentences to introduce the physical meaning of each law as suggested by the reviewer. The physical idea behind what we referred to as Cornford sliding law, is to combine Weertman and Columb behaviour in one sliding law. This is done by taking the reciprocal sum of basal drag as given by those

two sliding laws and raising each term to the power of m. This can be referred to as "rpCW", ie as the "reciprocal sum, raised to the power m, of Coulomb and Weertman sliding laws". We understand the comment raised by the reviewer and will simply refer to the sliding law as "rpCW". The equations defying the sliding law should make it clear why we use this abbreviation.

Eq 2: use the same conventions as eq 1 and eq 3 (\tau ^B_b to match \tau ^W -b) We will update this notation.

Eq 7 : looks incorrect, as though the second term inside the integral has been copy-pasted from the first and then not edited in some way. We will update this part of the equation.

L202: 'fully converged' : really? That is unusual, if not impossible. Was there a stopping criterion? We acknowledge that using the term 'fully' converged was inappropriate here and we will remove it. Instead, we mean that we stop the inversion when the amount that the misfit is changing with each step is very small. We will add in a statement about the stopping criteria for each inversion, in the form of the norm of the gradient.

**References**

Joughin I, Smith BE, Schoof CG. Regularized Coulomb Friction Laws for Ice Sheet Sliding: Application to Pine Island Glacier, Antarctica. Geophys Res Lett. 2019 May 16;46(9):4764-4771. doi: 10.1029/2019GL082526. Epub 2019 May 13. PMID: 31244498; PMCID: PMC6582595.

---

## Author Response (AR1)

**Reviewer 1**

Carr et al describe numerical simulations of three major Greenland glaciers from the present day to 2100. For each glacier an ensemble of simulations is constructed by varying (a) the climate (via the surface mass balance) according to published future projections, and (b) the physics approximation of sliding at the ice bed. All simulations begin with ice thickness and velocity in line with present day observations. They conclude that their variation in sliding physics has a far lower impact on future sea level rise than their variation in SMB. I would argue that this is model sensitivity rather than uncertainty, without observational calibration of the evolution, but the same language of uncertainty crops up elsewhere. Amended in the title, abstract and throughout the paper.

The experiments make sense as modelling studies, and the conclusions are justified (although differing from one paper, which is discussed) but an important sliding law has been omitted. This is the regularized Coulomb law (e.g Joughin 2019), which is related to the Schoof and Tsai rules that are discussed, but produces Coulomb-like sliding over a wider region of a typical glacier. It might make little difference, but should be considered. It is particularly important because it agrees most with time-dependent satellite observations in some regions (which is also worth noting in the introduction). We thank the reviewer for highlighting this law, which was very recently implemented in Ùa and was therefore not used in the original study. It would take a considerable amount of time to re-run all of our experiments with this additional law, so we feel that its inclusion is beyond the scope of the current paper. As the reviewer notes, as it may well make little difference: one of the key messages of our paper is that sliding law choice generally makes very little difference to sea level rise and we already include the Tsai law, so we do not expect significantly different results with the Joughin law. We note that the Joughin work is referenced in the introduction and we have added a note to the discussion that adding extra sliding laws in unlikely to have a significant impact on our conclusions (**Line 438**).

I understood the paper in general although there are a small number of typos/grammar errors.

**Specific comments**

Fig 3 is a good figure, but the labels are small. We have increased the label size, within the constraints of being able to fit the figure into one page width.

Figures 5,6,7,8 repeat the same data shown in Fig 4. It is not obvious to me that they serve a purpose. At the same time, other useful figures, e.g plots of the model initial state vs observations are not included. We have removed Fig 5 (the bar graph), as we agree it shows the same data as Fig 3 (the heat map). However, we have retained Figs 4 and 6-8 as these do show different data: i.e. the change in sea level rise contribution over time, versus the end value, as in Fig 3 (the heat map). We also see it as useful to retain Figs 6-8, as well as Fig 4, as they clearly illustrate that sliding law has very little impact on sea level rise contribution. We have kept the figures relating to the model set up etc in the supplementary information, so that those interested in the modelling aspects can look them up, whereas those with more general interest can understand our results from the main paper alone. However, if there are any of the setup figures the reviewer feels would be particularly useful to move to the main text, we can do so.

Please do not name a 'Cornford' sliding law (line 90, 130 and elsewhere). It appears in earlier work (which is cited in both the Cornford and Asay-Davies papers mentioned), and Cornford certainly makes no claim to it . It will also be useful to say something about the physical meaning of each law when they are introduced. We have changed the name of this law to 'modified Weertman-Coulomb' throughout and we have a brief description of each law in the sliding laws section (Section 3.1).

Eq 2: use the same conventions as eq 1 and eq 3 ($\tau^B_b$ to match $\tau^W$ -b) Updated.

Eq 7: looks incorrect, as though the second term inside the integral has been copy-pasted from the first and then not edited in some way. We have checked this equation and believe it is correct. However, if the reviewer can let us know exactly what they believe is incorrect, we can check again.

L202: 'fully converged' : really? That is unusual, if not impossible. Was there a stopping criterion? We acknowledge that using the term 'fully' converged was inappropriate here and we will remove it. Instead, we mean that we stop the inversion when the amount that the misfit is changing with each step is very small. We have updated the text on **Line 214** to state reflect this.

**References**

Joughin I, Smith BE, Schoof CG. Regularized Coulomb Friction Laws for Ice Sheet Sliding: Application to Pine Island Glacier, Antarctica. Geophys Res Lett. 2019 May 16;46(9):4764-4771. doi: 10.1029/2019GL082526. Epub 2019 May 13. PMID: 31244498; PMCID: PMC6582595.

**Reviewer 2**

This modeling study investigates the relative impact of a combination of four sliding laws and 12 future surface mass balance scenarios on mass loss from three major Greenland outlet glaciers through 2100. The four sliding laws are selected to include a variety of common modeling applications — these include sliding laws that prescribe the relationship between basal drag and sliding (Weertman and Budd), one where basal drag is directly prescribed (Tsai, in the manuscript), and a combination of the two (Cornford). The three glacier experiments are initialized with the same input data and model parameters. The authors find that the choice of SMB scenario imparts far greater variability on projected mass loss than choice of sliding law for all three study glaciers. By contrast, the SMB scenario used had minimal impact on the magnitude of grounding line retreat, which occurred for all three glaciers in the study. The authors note similar behavior from the two northern outlets, Petermann and Humboldt, with greatest mass losses projected using the Budd sliding law. Interestingly, Kangerdlugssuaq (SE Greenland) exhibited, on average, the smallest net mass losses, which were greatest when implementing the Weertman sliding law and an RCP 2.6 SMB scenario. In addition to suggesting a relatively minimized impact of sliding law choice compared to future SMB scenario, study results also emphasize the likely future importance of Humboldt for Greenland wide mass loss, as this catchment showed the highest SLR contribution of nearly 8mm, without considering additional mass losses from elevation feedbacks or dynamic thinning and terminus change.

This manuscript is clearly written and presents important glaciological work, as the impact of sliding choice is quite understudied for Greenland outlet glaciers. The results will add value to the community and add insight to the relative importance of remaining unknowns to future mass loss scenarios. The methods are overall sound and well justified, and clearly described in the main text and supplement. With minor revisions, this manuscript will be suitable for publication in TC. Some figures may be condensed or combined if manuscript length is of concern. I have included my main comments and requests for clarification first, followed by a few minor comments and edits.

Comments and requests for clarification

I understand the motivation for holding terminus fronts static and neglecting elevation/dynamic thinning feedbacks for these experiments, given the focus on SMB+sliding law and high computational load. However, I wonder how these exclusions impact the interpretation of grounding line retreat, given that HU and KG, assumed to terminate at the grounding line (no or negligible floating extensions), both show retreated grounding lines while holding initial 2015 fronts stationary. Wouldn't this create an artificial floating ice tongue by the end of the simulations? In the discussion, reduced SLR projections stemming from the Budd sliding law

at KG s attributed to enhanced sensitivity to changing ice thickness. This is suggested to indicate that the use of the Budd sliding law results in unrealistic SLR projections. Given the sensitivity to effective pressures and related basal shear stress in implementing Budd sliding law, does it not also seem reasonable that for KG, where some SMB scenarios resulting in net thickening (SMB gains), this would then manifest in reduced mass losses in Budd sliding law relative to the other sliding laws? In response to the question about the calving fronts, we prescribe a natural boundary condition and do not include calving: as the reviewer notes this is to separate out the impacts of SMB and sliding law and for computational efficiency. As noted by the reviewer, this does create a small floating section, but this section was always small and comparable to the width of HU / KG. We have noted this in the manuscript at **Line 164**. In future work, we plan to quantify the relative impact of a range of calving laws.

As noted by the reviewer, the Budd sliding law results in reduced mass losses compared to other sliding laws at KG. We note that the application of Budd sliding law, as used here and in several similar studies in that past, is somewhat questionable as it depends on the effective pressure changes throughout the whole computational domain, and not just in the vicinity of the grounding line. Despite our own reservations about this sliding law, we have nevertheless included it as it is a widely known sliding law and has been used in numerous previous studies.

I may have missed this reference in the text, but what is the cause of NaN values for several Weertman SMB runs for Humboldt Glacier (see Figure 3)? There are the runs with NaN, which we realised just prior to submission we needed to re-run. Theya re currently still running and we will add them to the next version of the manuscript – apologies for the delay on this.

Line 170 "Annual surface mass balance (SMB) was used to initialise the transient runs and was sourced from RACMO v2.3 (Noël et al., 2016). We used the average SMB for the years 2013-2017…" Can you please explain the use of RACMO for initialization, but use of MAR for downscaling CMIP5 and CMIP6 for the SMB scenarios? We used RACMO for the initialisation as we believed it was marginally better at representing contemporary Greenland surface mass balance and initially, we were only going to test the impact of sliding laws, not SMB forecasts. The downscaling of the CMIP5 and CMIP6 scenarios with MAR was done as part of another study (Hofer et al., 2020), and, importantly, was done for 13 different scenarios, which enabled us to test the impact of that range of SMB scenarios. Through the inversion and initialisation process, we ensured that our initial model state closely matched observed ice velocities and surface elevation change, so that we were confident our initial model set up matched well with observations at the start of the forward runs (i.e. 2015). Thus, any difference in initial model state due to the choice of SMB product is very unlikely to have impacted our results and would be very much smaller than the differences we observed between the various SMB forecasts. We have added a brief explanation to this effect on **Line 178.**

Line 214 "We subtracted the misfit between modelled and observed rates of thickness change (˙h), determined from Cryosat-2 two-year mean data for 2014-2015 (Simonsen and Sørensen, 2017)…" Does this indicate that the misfit was computed only at a single time step (here the 2 year mean thickness change during 2014-2015)? Or that this 2-year period serves as a reference elevation for deriving thickness changes? If the latter, can you clarify the period over which dh/dt relative to the reference period were computed? 2010-2015? Yes, this is correct that we use a single timestamp of thickness change: the 2-year mean thickness change during 2014-2015 serves as a reference. We have updated the text to clarify this at **Line 227**.

Line 252 through 255 Figure references indicate 3b showing Pg results, and Figure 3c showing results from KG. I believe these need to be swapped. We thank the reviewer for spotting this error and have updated it throughout.

On BedMachine v3 Why do the authors elect to use BedMachine version 3 over a newer version? Is the topography similar at the three study catchments for version 3 and the newer version of BedMachine? We selected BedMachine v3 because BedMachine 4 was not

available at the time when we conducted the model runs. The topography for the study glaciers is similar between the versions and it would take months to re-run the experiments for BedMachine4, so this is unfortunately beyond the scope of this paper. In future work, it would be very interesting to assess the impact of different bed products and/or different versions of the bed. However, we are confident that the swap to BedMachine v4 alone is unlikely to result in major differences in our results.

Line 296 Please double-check to ensure figure references and subplot references correspond to the correct glacier. We thank the reviewer for picking up this error and we have corrected it throughout the results and discussion.

Suggestions for Figures 6—8 Consider using a combination of solid, dotted, and dashed lines for figures that are stacked in a way that all curves appear in the figure. As shown, only W-N0 and rCW-N0 are readable in many subplots, and it is unclear which of these scenarios are overlapping the other two. Updated. We note that it is not possible in some cases for all lines to appear in each figure, as they directly overlap at the plot scale. However, we believe this underscores our key point in the paper, which is that sliding law has a limited impact on sea level rise values.

Minor Comments

Consider adding labels to subplots in Figure 3 if they are referenced by alphabetic notation in the text ("Figure 3c" for example). Thank you for noticing this – we have added the alphabetic notation.

Table 1 – Consider adding in paratheses what percentage of mean basin velocity is represented by the misfit value. We considered this but believe that it overly complicates the table. We would also note that the values given in Table 1 are actually the area-integrated, average misfit, as defined in equations 6 & 7, so is not directly comparable to mean basin velocities. We have corrected the table caption to correct this error.

Figure 4 – consider an alternate color ramp or alternating line styles and thicknesses. The yellow and light green colors are challenging to see and discern from one another in the figures. Amended – we have changed the line colours and styles in this figure and hope that it is now clearer to the reader.